# Genetic Manipulation of CB1 Cannabinoid Receptors Reveals a Role in Maintaining Proper Skeletal Muscle Morphology and Function in Mice

**DOI:** 10.3390/ijms232415653

**Published:** 2022-12-09

**Authors:** Zoltán Singlár, Nyamkhuu Ganbat, Péter Szentesi, Nomin Osgonsandag, László Szabó, Andrea Telek, János Fodor, Beatrix Dienes, Mónika Gönczi, László Csernoch, Mónika Sztretye

**Affiliations:** 1Department of Physiology, Faculty of Medicine, University of Debrecen, 4012 Debrecen, Hungary; 2Doctoral School of Molecular Medicine, University of Debrecen, 4012 Debrecen, Hungary; 3Cell Physiology Research Group, Eötvös Loránd Research Network (ELKH), 4012 Debrecen, Hungary

**Keywords:** endocannabinoid system (ECS), cannabinoid receptor of type 1 (CBR1), skeletal muscle force, contractility, excitation-contraction coupling (ECC), mitochondria, intracellular calcium

## Abstract

The endocannabinoid system (ECS) refers to a widespread signaling system and its alteration is implicated in a growing number of human diseases. Cannabinoid receptors (CBRs) are highly expressed in the central nervous system and many peripheral tissues. Evidence suggests that CB1Rs are expressed in human and murine skeletal muscle mainly in the cell membrane, but a subpopulation is present also in the mitochondria. However, very little is known about the latter population. To date, the connection between the function of CB1Rs and the regulation of intracellular Ca^2+^ signaling has not been investigated yet. Tamoxifen-inducible skeletal muscle-specific conditional CB1 knock-down (skmCB1-KD, hereafter referred to as Cre^+/−^) mice were used in this study for functional and morphological analysis. After confirming CB1R down-regulation on the mRNA and protein level, we performed in vitro muscle force measurements and found that peak twitch, tetanus, and fatigue were decreased significantly in Cre^+/−^ mice. Resting intracellular calcium concentration, voltage dependence of the calcium transients as well as the activity dependent mitochondrial calcium uptake were essentially unaltered by Cnr1 gene manipulation. Nevertheless, we found striking differences in the ultrastructural architecture of the mitochondrial network of muscle tissue from the Cre^+/−^ mice. Our results suggest a role of CB1Rs in maintaining physiological muscle function and morphology. Targeting ECS could be a potential tool in certain diseases, including muscular dystrophies where increased endocannabinoid levels have already been described.

## 1. Introduction

The endocannabinoid system (ECS) is a broadly distributed network that regulates a myriad of physiological processes such as appetite, body weight, energy homeostasis, mood, pain management, neuroprotection, muscle contractility, inflammation, immune and cognitive functions, just to name a few [1,2,3,4,5]. Its alteration is implicated in a growing number of human diseases [6,7,8,9,10]; thus, to better understand the underlying mechanisms, it is justified to decipher the function and dysfunction of the ECS under these conditions. Most of the aforementioned effects are mediated via G protein-coupled receptors named cannabinoid receptors (CBRs) [11,12,13] ubiquitously expressed in the nervous (CB1R) and immune system (CB2R). Besides their central localization, CB1Rs are widely distributed in peripheral tissues (e.g., in human and murine skeletal muscle) where they are localized mainly in the cell membrane (pCB1R) [6,14,15,16,17], but a subpopulation is present in the mitochondria (mtCB1R) as well. Here they are involved in cellular functions and energy metabolism [13,18,19,20,21] exerting several biochemical effects, including ATP production, modulation of reactive oxygen species (ROS), and neuropeptide signaling [22].

The process where skeletal muscle excitation, arriving in the form of an action potential initiated by moto neuron firing, is translated to a cytoplasmic Ca^2+^ signal that activates muscle contraction is termed excitation–contraction coupling (ECC) [23,24,25]. Altered ECC function results in a modified supply of Ca^2+^ ions to the contractile elements (directly, or indirectly, e.g., through mitochondrial influences), thus it has an impact on muscle performance overall. In skeletal muscle, mitochondria occupy about 10–15% of the muscle fiber volume [26], and are a major source for ATP and ROS production. Over the years, emerging evidence has supported the fact that mitochondria also serve as large and dynamic physiological buffers for Ca^2+^ [27]. Thus, it is now accepted that mitochondria, these highly dynamic organelles that constantly undergo fission and fusion, actively contribute to the spatio–temporal distribution of intracellular calcium concentrations [28,29].

We demonstrated earlier that CB1Rs are involved in the regulation of Ca^2+^ homeostasis of the skeletal muscle via a G_i_ protein and through a PKA-mediated mechanism [30]. The activation of mtCB1Rs in muscle cells is associated with the mitochondrial regulation of oxidative activity [31]. To date, the connection between the function of CB1Rs and the regulation of Ca^2+^ signaling has not been investigated yet. For this reason, here we developed and characterized a skeletal muscle-specific CB1R knock-down mouse model (Cre^+/−^) and we sought to investigate the impact of Cnr1 gene ablation on muscle performance and calcium homeostasis. Our findings suggest a role of CB1Rs and thus cannabinoid signaling in maintaining proper muscle performance and mitochondrial morphology. Under our experimental conditions, skeletal muscle-specific Cnr1 genetic ablation led to altered force production and hindered mitochondrial network ultrastructure without significant alterations in calcium signaling.

## 2. Results

### 2.1. Skeletal Muscle Specific Knock-Down of CB1R Resulted in Maintained Specimen Phenotype

To assess the function of CB1R in skeletal muscles, we generated a mouse model with Tamoxifen-inducible skeletal muscle-specific knock-down of Cnr1 gene using the Cre/LoxP system (Figure 1A). Briefly, mice containing two floxed CB1 alleles (CB1^flox/flox^) were crossed with mice containing the Cre recombinase cassette MerCreMer [MCM, mutated estrogen receptor (Mer) fused on both the NH_2_ and COOH terminal of Cre] expressed under the control of skeletal muscle-specific promoter [human α-skeletal actin (HSA)]. The MCM complex dissociates following the administration of Tamoxifen (2 months’ diet in our experiments) so that Cre recombinase will catalyze the excision of the floxed genomic DNA segment on Cnr1 gene. The additional frame shift mutation downstream to this exon results in an inability to produce mature mRNAs in the skeletal muscles of the floxed Cre^+/−^ mice. In Cre^−/−^ mice, the original Cnr1 gene function remains untouched upon Tamoxifen administration.

The genetic differentiation of loxP genotypes and the presence of the Cre construct were identified via gel electrophoresis (Figure 1B). Following 2 months of Tamoxifen feeding, mixed-gender animal groups were selected for the in vivo and in vitro experiments (Figure 1C). To estimate the muscle-specific downregulation of CB1R mRNA expression, total lysates of *m. extensor digitorum longus* (EDL), *m. soleus* (SOL), and *m. tibialis anterior* (TA) were examined and a significant partial downregulation of CB1R mRNA levels (to 34%, 19%, and 34% respectively, compared to similar EDL, SOL and TA samples originated from Cre^−/−^ mice, Figure 1D) was detected with qPCR. In the Cre^+/−^ samples, the downregulation of CB1 mRNA levels was not significantly different in the three muscle types examined here. The qPCR investigation was performed in triplicates and similar results were reached using Ap3D1 as an internal control. The significant reduction in CB1R mRNA level correlated with a modest but still significant reduction of CB1 protein level (~23% average reduction compared to Cre^−/−^) based on the semi-quantitative Western Blot method analysis of whole muscle homogenate samples performed on TA muscles from six animals in each group (Figure 1E,F). Furthermore, the analysis revealed that the genetic manipulation of Cnr1 did not induce an alteration of the CB2R protein expression level (Appendix A).

At 3 months, no changes in terms of phenotype and body weight gain were observed neither at Cre^−/−^ nor at Cre^+ /−^ mice. (Figure 1C,G). To check the in vivo muscle performance of the animals, grip tests were used at the beginning (1 month of age) and at the end (3 months of age) of the special Tamoxifen diet period examining four mice in each group. The maximal grip force normalized to body weight decreased significantly following 2 months’ Tamoxifen diet (Figure 1H) in Cre^+/−^ mice.

### 2.2. Skeletal Muscle Specific Knock-Down of CB1R Resulted in Altered Isometric Force

To determine whether Cnr1 genetic manipulation (besides the significant influence on muscle grip force measured in vivo) has any effect on muscle function, in vitro muscle strength was investigated on fast-twitch EDL and slow-twitch SOL muscles. There was no significant difference between Cre^−/−^ and Cre^+/−^ mice in the mean amplitude of the normalized single twitches of EDL muscles (Figure 2A,E and Table 1). However, for the SOL muscles, these values were significantly altered (Figure 2B,G and Table 1).

On the other hand, tetanic force was drastically decreased upon Cnr1 down-regulation in both investigated muscle types (Figure 2C,D,F,H and Table 1).

Interestingly, the twitches did not change (only TTP in SOL), but almost all kinetic parameters of tetani were altered significantly following Cnr1 genetic manipulation in Cre^+/−^ mice. An exemption was the duration in EDL and TTP in SOL. Furthermore, the fatigability and the force frequency relationship changed dramatically in Cre^+/−^ mice (Figure 3). Both types of muscle showed faster fatigue and had decreased tolerance towards high-frequency stimulation. When analyzing the fatigue, one could clearly see that in Cre^+/−^ EDL muscles the fatigue was presenting a significantly faster decrease compared to Cre^−/−^ for a longer interval than in SOL muscles; however, towards the end of the stimulation protocol (e.g., after 100 tetani), this difference vanished and was no longer present (Figure 3A,B). The force-frequency analysis revealed that, at low frequency stimulation, no differences could be seen between the two samples for both muscle types; however, when applying high frequencies (above 100 Hz in EDL and 60 Hz in SOL, respectively) upon reaching the peak, a marked decline in tetanic force was observed in the Cre^+/−^ muscles (Figure 3C,D).

Taken together, the in vivo and in vitro force measurements suggest that CB1 fundamentally contributes to the normal skeletal muscle force production and performance in mice.

### 2.3. Skeletal Muscle Specific Knock-Down of CB1R Resulted in Unaffected Resting Intracellular Calcium Levels and Release Channel Sensitivity to Activation

To study whether Cnr1 genetic manipulation has any effect on the coupling between DHPR and RyR1 in skeletal muscles, we recorded calcium transients evoked by 100 ms long rectangular depolarizations between –60 mV and +30 mV on enzymatically dissociated single *m. flexor digitorum brevis* (FDB) fibers. The whole-cell voltage-clamp technique was used to elicit calcium transients in conjunction with confocal microscopy imaging. Figure 4A,B display two representative line-scan images from a Cre^−/−^ and a Cre^+/−^ FDB fiber. Figure 4C summarizes the voltage dependence of the normalized fluorescence from 21 Cre^−/−^ and 27 Cre^+/−^ fibers obtained from independent experiments similar to those shown in Figure 4A,B.

When comparing data obtained from Cre^−/−^ fibers versus Cre^+/−^ fibers, neither the voltages of half-maximal activation (V_50_) nor the slopes of the fitted curves (*k*) (Figure 4C), and the calculated *Ca_max_* values showed any differences (Appendix A). The mean values of peak fluorescence at maximal activation (+30 mV) were also not statistically different between the two groups (Figure 4D). Furthermore, when examining the resting intracellular calcium levels on Fura2-AM loaded FDB cells (Figure 4E), we found these to be essentially identical, with no statistical difference.

Taken together, these data suggest that under the above-described experimental approach and employed stimulation protocols, intracellular Ca^2+^ handling and activation of Ca^2+^ release was similar in the two samples. Consequently, we draw the conclusion that Cnr1 genetic manipulation does not seem to alter the ECC machinery and has no significant impact on the resting calcium concentration and on release channel activation in FDB muscle cells under our experimental conditions. These findings imply that the smaller tetanic force observed in Cre^+/−^ mice is not the consequence of modified calcium handling and release channel activity.

### 2.4. Skeletal Muscle Specific Knock-Down of CB1R Resulted in Unimpaired Fatigability of Ca^2+^ Release

We next set out to examine if, upon activation, the replenishment of the SR could be modified by the CB1 genetic manipulation in mice. To test this, a series of tetani with the pattern illustrated in Figure 5A,B mimicking maximal depolarizations (+30 mV, 200 ms) close to the train of action potentials that activate Ca^2+^ release in the living animal were applied to FDB fibers from both Cre^−/−^ (*n* = 19 fibers, *N* = 6 mice, Figure 5A) and Cre^+/−^ mice (*n* = 26 fibers, *N* = 12 mice, Figure 5B). These calcium transients obtained with the continuous repeated depolarizations can be considered similar to physiological tetanic trains during muscle contraction.

Ca^2+^ release flux was derived from the calculated cytosolic calcium (*[Ca^2+^]_cyto_*) which were eventually determined from the normalized fluorescence values averaged over the spatial domain: F(*t*)/F_0_ (white traces in Figure 5A,B). Figure 5C depicts two representative examples of the total amount of calcium released upon repetitive stimulation obtained as the time integrals of the release flux presented in Figure 5D. A simplified approach of the removal method [32,33,34] was used to calculate the release flux (Figure 5D). The release flux presents normal features as highlighted on the inset of Figure 5D: (i) a “peak” during which the flux quickly reached its maximum, and (ii) a “steady state” (S) that flattens within a few tens of milliseconds and is maintained during the course of the depolarization. Appendix A summarizes the average fluorescence values and the parameters of the release fluxes calculated for both samples. To estimate the average amount of calcium in the SR from the depolarizing train, single-exponential functions were fitted to the eight points of the series (Figure 5E). The exponential decay was assessed by using Equation (2). The overlap of the exponential decay curves illustrating the calcium decline following tetanic stimulation suggest a similar propensity for depletion in the two samples that is essentially unaltered by Cnr1 genetic manipulation.

### 2.5. Skeletal Muscle Specific Knock-Down of CB1R Resulted in Unaltered Activity Dependent Mitochondrial Calcium Uptake

Since a subpopulation of CB1 is localized in the (outer) mitochondrial membrane, we explored if its down-regulation could potentially alter the mitochondrial calcium homeostasis and the process of calcium uptake. This idea was further supported by the decreased calcium-sensitive regulator of the mitochondrial uniporter 1 (MICU1) protein expression levels confirmed by Western Blot analysis of TA muscles from the two animal groups (Appendix A). Figure 6A–C show representative confocal xy images of Rhod-2 fluorescence in a Cre^+/−^ FDB fiber at rest (Figure 6A), following one (Figure 6B) and five (Figure 6C) consecutive tetanic depolarizing pulses to supramaximal voltages applied via platinum electrodes placed in close proximity to the FDB muscle cell of interest. The fluorescence (F) averaged over the spatial domain (x) illustrates that, as expected, Rhod-2 fluorescence that monitors the activity-dependent mitochondrial calcium accumulation increased modestly upon tetanic stimulation (Figure 6D–F) in both Cre^−/−^ and Cre^+/−^ samples. However, no statistical difference was found when comparing average fluorescence increase (*F_mito_*) (Figure 6G). Thus, we concluded that it is not an altered mitochondrial calcium handling that accounts for the smaller tetanic force observed in Cre^+/−^ muscle fibers. Consequently, another underlying mechanism must be at work and be responsible for this phenomenon.

### 2.6. Skeletal Muscle Specific Knock-Down of CB1R Resulted in Significant Alterations of Mitochondrial Morphology

To describe possible structural changes within the myofibrillar network in adult skeletal muscle fibers as a consequence of the reduced CB1 expression, electron microscopy (EM) was performed on TA muscles. EM images were taken on longitudinal muscle sections prepared from Cre^−/−^ and Cre^+/−^ animals and two representative images are shown in Figure 7A,B. To characterize the changes in morphology, we have quantified the area (Figure 7C), perimeter (Figure 7D), roundness (Figure 7E), aspect ratio (Figure 7F), and form factor (Figure 7G) for each identified mitochondrion in all EM micrographs. Significant changes in samples originated from Cre^+/−^ were found in almost all the examined parameters. The relative distribution of areas (Figure 7H) were found to be significantly increased as well in Cre^+/−^ samples compared to Cre^−/−^ animals. This observation is further exemplified in the inset of Figure 7H, where a clearly noticeable rightward shift of the area distribution is illustrated highlighting the appearance of mitochondria with large areas in Cre^+/−^ muscles, a phenomenon not observed in Cre^−/−^ samples. Similarly, significantly smaller perimeter and area were obtained in Cre^+/−^ fibers when analyzing cross-sectional EM images of TA muscles.

These results suggest that mitochondrial morphology, but not the activity-dependent mitochondrial calcium uptake, was severely affected by the alteration of CB1 expression in skeletal muscles.

## 3. Discussion

Over-activity of the ECS and alteration of the cannabinoid tone had been described in a large number of disorders, including muscle diseases such as Duchenne Muscular Dystrophy (DMD) [35], obesity [36,37] and aging [38]. Hence, CB1 antagonism has been proposed as a promising target to treat cachexia and sarcopenia through modulation of the metabolism and muscle regenerative capacity [39], and more recently as an add-on therapy in DMD which could decrease the usage of steroidal anti-inflammatory drugs, presenting serious side effects. Besides the overacting ECS, disturbances of Ca^2+^ homeostasis can also severely impact muscle physiology and proper function [40]. Therefore, the function and dysfunction of the ECS in muscle is an important focus of research interest to better understand the underlying mechanisms of ECS-related disorders.

Previous studies revealed CB1R expression in murine and human skeletal muscles [31], but the relevance, if any, of muscle CB1Rs in muscle biology has remained elusive. Nevertheless, CB1 has been described as a negative regulator of oxidative metabolism, muscle anabolism, satellite cell growth and muscle regeneration [38].

The use of systemic and peripheral CB1 knock-out mice has been implemented for a while and there are more and more studies using genetically modified specimens [35,36,41,42]. Our workgroup had previously shown interest in studying endocannabinoid signaling in skeletal muscle and we demonstrated that sarcoplasmic Ca^2+^ release in adult and cultured mammalian skeletal muscles is affected and regulated by the ECS [30]. However, the study was performed on a systemic CB1 knock-out mouse model; thus, the obtained data raised the question of whether the effect of the cannabinoids on the muscle occurs directly, or indirectly by modulating the nervous control. In the present work, we sought to generate a skeletal muscle-specific Tamoxifen-inducible CB1 knock-down mouse model and we aimed to characterize this animal model to determine the functional role of CB1Rs and their impact on muscle performance and morphology in young adult mice.

Muscle type and age might be an important determinant of CB1 expression. However, there are equivocal studies on this, as CB1 expression was reported to be lower in the *m. gastrocnemius* of 1 year vs. 2 weeks old mice [43], while others reported a higher CB1 expression in the *m. gastrocnemius* of 16-month old vs. 4-month old mice [44]. In human muscle tissues, CB1 expression was found to be higher in slow-twitch type I fibers (seen in high abundance in elite endurance athletes, such as long-distance runners and cyclists) compared to fast-twitch IIa and IIx fibers (abundant in elite power athletes, such as weightlifters and sprinters) [39]. Likewise in mice, CB1 protein expression was described as rather conflicting: some laboratories found higher CB1 protein expression in predominantly fast (e.g., *gastrocnemius*, *tibialis anterior*) vs. slow (e.g., *soleus*) muscles [43], while others reported the opposite [38]. Our present results are in agreement with the latter, which from a metabolic perspective seems indeed more logical: CB1 is known to be an important regulatory player in peripheral oxidative capacity; thus, one would expect it to be more abundant in type I fibers, which exhibit a higher oxidative capacity than type II fibers. Our results reveal that genetic manipulation of Cnr1 in skeletal muscles led to decreased in vivo grip force and in vitro tetanic force production in both fast- and slow-twitch muscles. We demonstrated that the Cre^+/−^ mice are more prone to fatigue upon long-lasting tetanic stimulation under in vitro circumstances (Figure 3A,B). The different expression levels of the CB1 protein could be an explanation as to why our in vitro force measurements revealed significant differences between the two samples on EDL muscles for only the tetanic stimulations, whereas for SOL. the significant differences could already be detected for the twitch stimulations. Altogether, these findings are in line with previous studies on CB1 research in skeletal muscle [30] where CB1 signaling has been described to have negative effects on the expression of genes regulating oxidation and insulin sensitivity [6,45], the glucose/pyruvate/lactate pathways, and their effects additionally on mitochondrial energetics [8].

The distribution of CB1 receptors may also depend on the quantity of muscle mitochondria, which are highly dynamic organelles and continuously change their size and shape in response to muscle activity. Moreover, cellular Ca^2+^ dynamics are known to differ considerably across fiber types both in amplitude and in frequency [46,47], which likely expose mitochondria to different levels of Ca^2+^. Mendizabal-Zubiega and colleagues [31] elegantly showed that the majority of CB1 receptors in skeletal muscles (*gastrocnemius* and *rectus abdomini*, both enriched in type I slow-twitch fibers just like the *soleus* muscle used by us) were present on mitochondrial membranes (55 and 78% of the total immunoparticles) as opposed to the brain, where they are preferentially located in neuronal membranes [13]. In our studies, we have analyzed various muscle types with different mitochondrial content: (i) fast-twitch muscles (EDL and FDB) that are less abundant in mitochondria, enriched mainly in type II glycolytic fibers; (ii) SOL muscles which contain predominantly type I slow-twitch oxidative fibers and are abundant in mitochondria [48]; (iii) mixed muscles such as TA, which were employed for protein content and morphological characterization. Interestingly, in the glycolytic EDL muscle, which requires higher energy and increased ATP turnover to sustain the prolonged muscle contraction under physiological conditions, upon tetanic stimulation, decreased specific force production was observed (Figure 2C,F). A possible interpretation for this could be due to mitochondria within glycolytic fibers relying on two shuttles (e.g., glycerophosphate and malate-aspartate shuttles) to import cytosolic-reducing equivalents, compared with only one (e.g., malate-aspartate shuttle) in mitochondria from slow-oxidative fibers [49].

In our hands, the genetic ablation of Cnr1 induced dramatic morphological changes in the mitochondrial network as revealed by the EM studies (Figure 7). Nevertheless, these morphological alterations were not in line with the functional modification of the calcium fluxes as the process of SR calcium release and mitochondrial calcium uptake were not affected in the Cre^+/−^ mice. It is well known that the uptake is mediated via the calcium sensitive regulator of the mitochondrial uniporter (MICU1), which in turn has been described to modify ATP production by directly increasing the activity of certain metabolic enzymes via calcium [50]. However, a rather conflicting finding in the present work was the slight but significant reduction of MICU1 protein levels in the Cre^+/−^ mice (Appendix A) which may at least partially explain the reduced force generation capabilities in Cre^+/−^ animals. However, reduced ATP production would certainly alter SERCA pump function, which in turn would affect the intracellular calcium concentration and calcium fluxes across cellular membranes, a phenomenon we did not observe in our experiments. Furthermore, one cannot exclude the interplay between CB1 and ROS production as there is growing number of evidence that the ECS may play an important role in the regulation of cellular redox homeostasis [51,52]. Furthermore, activation of the CBR1s can potentially drive several different signaling pathways within muscle cells, depending on factors that are tissue and cell type specific. These pathways downstream of CB1R activation may include cAMP/PKA, MAPK/ERK and PI3K/AKT [53]. Studies have highlighted the role of CB1 cannabinoid receptors on GABAergic neurons in brain aging [54]. In skeletal muscle, Cnr1 genetic ablation could lead to a similar phenotype as in muscle aging or atrophy [55], where altered mitochondrial morphology was observed similarly to what we describe here.

Nevertheless, exercise training was proposed to induce muscle fiber type switching [56,57] and to upregulate circulating endocannabinoid levels (particularly AEA but also 2-AG); more specifically, 12 weeks of resistance exercise increased CB1 expression in human muscles [39], which the authors hypothesized to be caused by a change in the muscle fiber type composition. However, it was an intriguing finding that instead of an increase in type II fibers, which would have resulted in a relative decrease in CB1 expression, the authors found elevated CB1 expression levels without any muscle myosin heavy chain type I and II expression alteration due to resistance exercise.

Based on the above elaborated details, we hypothesize that the hindered force production might imply a potential role for CB1 receptors in the regulation of the oxidative activity of mitochondria, ATP and possibly ROS production, through the relevant enzymes implicated in the pyruvate metabolism, a main substrate for Krebs cycle activity (Figure 8). Furthermore, since calcium signaling was not seemingly altered upon Cnr1 genetic manipulation, a hindered contractile machinery activity and/or muscle fiber type composition could also be the reason behind the altered force production seen both in vivo and ex vivo. Nevertheless, further research is necessary to explore this and what may be the link between these players and ultimately to shed light on the molecular mechanisms underlying the downregulation of MICU1 due to the genetic ablation of Cnr1.

## 4. Materials and Methods

### 4.1. Animal Care and Generation of the Muscle Specific CB1 Knock-Down Mouse Strain

Animal experiments adhered to the guidelines of the European Community (86/609/EEC) and were designed to minimize animal suffering and distress. The experimental protocol was approved by the institutional Animal Care Committee of the University of Debrecen (3–1/2019/DEMAB). The mice were housed in plastic cages with mesh covers and fed ad libitum with pelleted mouse chow and water. Room illumination was an automated cycle of 12 h light and 12 h dark, and room temperature was maintained within the range 22–25 °C.

Tamoxifen-inducible, muscle-specific CB1 knock-down mice were obtained similarly as described in Gönczi et al. [58]. In short, we generated a mouse model that permit inducible Cnr1 ablation in adult skeletal muscle by crossing B6.Cg-Tg(ACTA1-Cre)79Jme/J mice (Jackson Laboratory, Bar Harbor, ME, USA) with C57BL/6-CB1^flox/flox^ mice (a kind gift from Prof. Dr. Beat Lutz and Matthias Gaestel (Institute of Physiological Chemistry University Medical Center of the Johannes Gutenberg University, Mainz, Germany). Mice hemizygous for this HSA-Cre transgene are viable, fertile, normal in size, and do not display any gross physical or behavioral deficiencies. The HSA-Cre transgenic mice have the Cre recombinase gene driven by the human alpha-skeletal actin (HSA or ACTA1) promoter. The HSA-MerCreMer construct encodes the Cre recombinase and contains a mutant estrogen ligand binding domain that requires the presence of Tamoxifen for activity. When bred with mice containing a loxP-flanked sequence of interest, Cre-mediated recombination will result in striated muscle-specific deletion of the flanked genome leading to Cre^+/−^-CB1^flox/flox^ Tamoxifen-inducible muscle-specific CB1 knock-down mice (hereinafter referred to as Cre^+/−^) and Cre^−/−^-CB1^flox/flox^ mice that did not have the MerCreMer construct as littermate controls (Cre^−/−^ in the following) (see also the breeding strategy illustrated in Figure 1A).

### 4.2. Tamoxifen Diet

To induce CB1 ablation Tamoxifen diet (per os) was started immediately after separation of the Cre^+/−^ and Cre^−/−^ pups from the mother at the age of 4 weeks. Littermates were fed for 2 months without interruption (while monitoring normal weight gain) and then were used for the subsequent experiments. The Tamoxifen-supplemented chow (Envigo, TD 130857) contained 500 mg Tamoxifen/kg diet, providing 80 mg Tamoxifen/kg body weight per day assuming 20–25 g body weight and 3–4 g daily food intake [59,60].

### 4.3. Molecular Biology

#### 4.3.1. Genotyping

Total genomic DNA were isolated from finger biopsies and screened for the presence of the HSA-Cre recombinase cassette by PCR (Biometra Tadvanced Twin 48 G, 230 V, Analytik Jena GmbH, Jena, Germany) using the appropriate primers spanned the C-terminus MerCre junction (see Table 2) and produced a 717-bp product.

For the floxed CB1 alleles, special primers were used for PCR to differentiate homozygous CB1^flox/flox^ mice (500-bp product) from wild type CB1^wt/wt^ (400-bp product) and heterozygous CB1^flox/wt^ mice (500 and 400-bp products) (Figure 1B). Genotype was examined using agarose gel electrophoresis.

#### 4.3.2. Quantitative PCR Analysis

Total ribonucleic acid (RNA) fractions were isolated using TRI reagent (MRC, Cincinnati, OH, USA, cat. no.: TR118) from homogenized *m. flexor digitorum brevis* (FDB), *m. extensor digitorum longus* (EDL), *m. soleus* (SOL), and *m. tibialis anterior* (TA) skeletal muscle specimens from Tamoxifen fed Cre^+/−^ and Cre^−/−^ mice. The isolated RNA samples were dissolved in nuclease-free water (NFW) and stored at −80 °C until further use. The RNA yield and purity were determined by a spectrophotometer at 260 nm wavelength (NanoDrop ND1000; Promega Biosciences, Madison, WI, USA). The isolated RNA samples were treated with DNase and RNase inhibitor (Ambion, Austin, TX, USA), then reverse transcription was applied using a High-Capacity cDNA Reverse Transcription Kit (Thermo Fisher, Waltham, MA, USA; cat. no.: 00735667). According to the manufacturer’s protocol, 500 ng of the isolated total RNAs were reverse transcribed into complementary DNA (cDNA). cDNA synthesis was carried out using random hexamers in 25 µL reaction volume. For quantitative RT-PCR, Taqman Gene Expression Assays were used with Taqman™ Gene Expression Master Mix (Applied Biosystems, Foster City, CA, USA). The amplification was performed using a Light Cycler 480 Master instrument (Roche, Basel, Switzerland) (cat. no. for plates, Roche: 04729692001; cat. no. for sealing foils, Roche: 04729757001). Mouse CB1R Taqman gene expression assays were purchased from Thermo Fisher Scientific (Waltham, MA, USA, Mm01212171_s1). The cycling conditions were 10 min at 95 °C, followed by 50 cycles of 15 s at 95 °C and 1 min at 60 °C. The relative expression values for each transcript of interest were calculated by the comparative C_t_ method. Each sample was run in triplicates and standardized to their own internal PPIA (Mm02342430_g1) and Ap3D1 (Mm00475961_m1) gene expression. The obtained mean values were then used for further analysis.

#### 4.3.3. Western Blot Analysis

TA skeletal muscle tissues were homogenized in lysis buffer (20 mM Tris–HCl, 5 mM EGTA, Protease Inhibitor Cocktail (Sigma, Saint Louis, MA, USA) with a HT Mini homogenizer (OPS Diagnostics, Lebanon, NJ, USA). Sixfold concentrated electrophoresis sample buffer (20 mM Tris–HCl, pH 7.4, 0.01% bromophenol blue dissolved in 10% SDS, 100 mM β-mercaptoethanol) was added to total lysates to adjust the equal protein concentration of samples, and then the mixture was boiled for 5 min at 90 °C. Next, 20 μg of total protein was loaded to each lane and separated on a 10% SDS–polyacrylamide gel. Proteins were transferred to nitrocellulose membranes, blocked with 5% non-fat milk dissolved in phosphate saline buffer (PBS), then membranes were incubated with the appropriate primary antibodies overnight at 4 °C. Anti-CB1 polyclonal antibody (Invitrogen, Waltham, Massachusetts, USA, PA5-85080, 1:100) anti MICU1 polyclonal antibody (Invitrogen, Waltham, Massachusetts, USA, PA5-77364), and anti-CB2 monoclonal antibody (Sigma-Aldrich, St. Louis, Missouri, USA, JB011-3C7) were used for specific labelling. After washing for 30 min in TBS supplemented with 1% Tween-20 (TBST), membranes were incubated with HRP-conjugated secondary antibodies (Blotting Grade Goat Anti-Rabbit IgG (H + L) (Human IgG Absorbed) Horseradish Peroxidase Conjugate (Bio-Rad, Hercules, California, USA, 170–6515) and Blotting Grade Affinity Purified Goat Anti-Mouse IgG (H + L) Horseradish Peroxidase Conjugate (170–6516). Membranes were developed and signals were detected using enhanced chemiluminescence (Thermo Fisher Scientific). The optical density of signals was measured by ImageJ software (NIH, Bethesda, MD, USA) and results were normalized to the optical density of alfa-actinin in the tissues (Santa Cruz Biotechnology, Dallas, TX, USA, sc-166524).

### 4.4. In Vivo Experiments

#### 4.4.1. Body Weight Measurement

The body weight of the mixed-gender mice was measured at the beginning of the Tamoxifen diet (week 4) and weekly afterwards until the end of the 2-month long feeding period (week 12) for each individual mouse in both the Cre^−/−^ and Cre^+/−^ group. The body weight increase was averaged by groups.

#### 4.4.2. Forepaw Grip Test

The force of forepaw was measured as described earlier [61]. Briefly, when the animals reliably grasped the bar of the grip test meter, they were then gently pulled away from the device by their tails. The maximal force before the animal released the bar was digitized at 2 kHz and stored by an online connected computer. The test was repeated 10–15 times on each animal to obtain a single data point. Grip test was assessed on the day when the mouse was sacrificed.

### 4.5. In Vitro Experiments

Mice were anaesthetized and sacrificed in compliance with the guidelines of the European Community (86/609/EEC). After pentobarbital anesthesia (27 mg/kg) and cervical dislocation, FDB, EDL, SOL, and TA from the hind limb were dissected manually under a transmitted light microscope using thin forceps and fine precision surgical scissors.

#### 4.5.1. Measurement of Muscle Force and Fatigue

Muscle contractions were measured as described in our previous reports [62,63]. In brief, fast- and slow-twitch muscles (EDL and SOL) were placed horizontally in an experimental chamber continuously superfused (10 mL/min) with Krebs’ solution (containing in mM: 135 NaCl, 5 KCl, 2.5 CaCl_2_, 1 MgSO_4_, 10 Hepes, 10 glucose, 10 NaHCO_3_; pH 7.2; room temperature) equilibrated with 95% O_2_ plus 5% CO_2_. One end of the muscle was attached to a rod and the other to a capacitive mechano-electric force transducer (Experimetria, Budapest, Hungary). Two platinum electrodes placed adjacent to the muscle were used to deliver short, supramaximal pulses of 2 ms in duration to elicit single twitches. Force responses were digitized at 2 kHz using Digidata 1200 A/D card and stored with Axotape software (Axon Instruments, Foster City, CA, USA). Muscles were then stretched by adjusting the position of the transducer to a length that produced the maximal force response and allowed to equilibrate for 6 min. Single pulses at 0.5 Hz were used to elicit single twitches. At least 10 twitches were measured under these conditions from every muscle. The individual force transients within such a train varied by less than 3% in amplitude; thus, the mean of the amplitude of all transients was used to characterize the given muscle. To elicit a tetanus, single pulses were applied with a frequency of 200 Hz for 200 ms (EDL) or 100 Hz for 500 ms (SOL).

Muscle fatigue was defined as the decline in isometric force development in repeated tetanic stimulations. In every 2 s (0.5 Hz), 150 consecutive tetanic contractions were elicited. The peak of all tetanic contractions was normalized to the first in the series. Durations of individual twitches and tetani were determined by calculating the time between the onset of the transient and the relaxation to 10% of maximal force.

The force–frequency relation in EDL was obtained with isometric contractions at 20, 40, 60, 80, 100, 120, 160 and 200 Hz, with trains of 200 ms duration, at 30 sec intervals. In SOL the protocol was 10, 20, 30, 40, 60, 80, and 100 Hz, with trains of 500 ms duration, at 30 s intervals.

#### 4.5.2. Transmission Electron Microscopy and Quantitative Analysis of EM Images

Freshly excised TA muscles were fixed in situ with fixative solution (3% glutaraldehyde in Millonig’s buffer). Small bundles of fixed muscle fibers were then postfixed in 1% OsO4 in water. For rapid dehydration of the specimens, graded ethanol followed by propylene-oxide intermediate was used. Samples were then embedded in Durcupan epoxy resin (Sigma). Ultrathin horizontal sections were cut using a Leica Ultracut UCT (Leica Microsystems, Wien, Austria) ultramicrotome and stained with uranyl acetate and lead citrate. Sections were examined with a JEM1010 transmission electron microscope (JEOL, Tokyo, Japan) equipped with an Olympus camera. Cross-sectional oriented EM micrographs were used at the same magnification (18,000×) and were analyzed using Image J software. The parameters of interest to describe the mitochondrial morphology were: area, perimeter, roundness, aspect ratio and form factor.

#### 4.5.3. Isolation of Single FDB Fibers

All calcium measurements were carried out on skeletal muscle fibers from the FDB muscle of the mouse. Calcium-free Ringer’s solution (containing in mM: 136 NaCl, 5 KCl, 1 MgCl_2_, 10 HEPES; 10 glucose; pH 7.2) was used during the dissection of the muscle. Single muscle fibers from FDB were enzymatically dissociated in minimal essential media containing 0.2% Type I collagenase (Sigma) at 37 °C for 45–50 min [64]. To release single fibers, the FDB muscles were mechanically dissociated / triturated gently in normal Ringer solution (same as above but supplemented with 2.6 mM CaCl_2_). The isolated fibers were then placed in culture dishes and stored at 4 °C in a refrigerator until use. Only fibers with clearly visible striations, no swelling or surface membrane damage were selected for experiments.

#### 4.5.4. Voltage Clamp, Confocal Microscopy, and Image Processing

The experimental design was similar to the one described by Sztretye et al. [63,65]. Briefly, isolated single FDB fibers were voltage-clamped (Axoclamp 2B, Axon Instruments) and imaged using a confocal microscope (Zeiss 5 Live, Oberkochen, Germany) equipped with a 20× air objective. Line-scan image recordings were synchronized with the application of depolarizing pulses via pClamp 11.0 (Molecular Devices, San Jose, CA, USA). The time resolution was 0.5 ms/line, while the space resolution was 0.24 µm/pixel.

The isolated fibers were dialyzed with the Rhod-2 containing internal solution though the patch pipette. Experimental temperature was 20–22 °C and the holding potential was −80 mV. Pipette resistance varied between 2 and 3 MΩ. The experiments were performed in the presence of 10 mM EGTA in the external solution, so that the endogenous buffers in the removal process were almost negligible. Correction for linear capacitive currents was performed by analog compensation.

Average cytosolic [Ca^2+^](*t*) was calculated from averaged Rhod-2 fluorescence F(*t*) as reported in Sztretye et al. [64].

The voltage dependence of activation was described by a Boltzmann function:(1)$$Ca\left(V\right)=\frac{Ca_{max}}{1+exp^{\frac{-\left(V_{m}-V_{50}\right)}{k}}}$$
to derive the transition voltage *V*_50_, limiting logarithmic slope 1/*k*, and *Ca_max_*.

Line-scan images were analyzed by an in-house custom-made program to obtain the fluorescence (F) and the background fluorescence (F_0_). From these, the program calculated the intracellular calcium values using the following parameters: K_d (Rhod-2)_ = 1.58 μM, k_ON_ = 0.07 µM^−1^ ms^−1^, and k_OFF_ = 130 s^−1^ [34] Individual data points were then normalized to Ca_max_ and plotted as a function of the membrane potential to display the voltage dependence of the activation. Peak F/F_0_ were obtained by averaging the data points in the spatial domain and plotted at close to maximal depolarization. When a series of eight tetanic depolarizations were studied, a single exponential function was used to fit the time course of the amount of calcium released during each pulse based on the following equation:(2)$$y=y_{0}+ae^{-bx}$$
where *x* is the number of tetanic pulses applied, *b* is the time constant of SR depletion and *a* is the remaining SR calcium content.

Ca^2+^ release flux was derived from [Ca^2+^](*t*) and subjected to a removal model fit analysis, which calculates release flux as that necessary to account for the evolution of [Ca^2+^]_i_(*t*) in a single compartment model that includes quantitatively specified processes of removal as originally described by Melzer et al. [32]. From the release flux (the flux exiting through the release channels) the net flux leaving the SR can be derived by subtraction of the pump removal flux. The integral of the net flux provides the SR content releasable by depolarization and otherwise termed the amount released.

External bath solution (in mM): 140 TEA-CH_3_SO_3_, 1 CaCl_2_, 3.5 MgCl_2_, 10 Hepes, 1 4-AP, 0.5 CdCl_2_, 0.3 LaCl_3_, 0.001 TTX (citrate), and 0.05 BTS (N-benzyl-p-toluene sulphonamide; Sigma-Aldrich). pH was adjusted to 7.2 with TEA-OH and osmolality was adjusted to 320 mOsm with TEA methanesulfonate.

Internal (pipette) solution (in mM): 110 N-methylglucamine, 110 L-glutamic acid, 10 EGTA, 10 Tris, 10 glucose, 5 Na ATP, 5 phosphocreatine Tris, 0.1 Rhod-2, 3.56 CaCl_2_, and 7.4 mM MgCl_2_ were added for a nominal 1 mM [Mg^2+^] and 100 nM [Ca^2+^]. pH was set to 7.2 with NaOH and osmolality to 320 mOsm with N-methylglucamine.

#### 4.5.5. Resting Myoplasmic [Ca^2+^]_i_ Measurement

The resting intracellular *Ca*^2+^ concentration ([*Ca*^2+^]_i_) was measured using Fura-2 AM fluorescent calcium indicator. Briefly, single-FDB fibers were mounted on a glass cover slip with 5 µM Fura-2 AM dye for 1 h. Fibers were then washed with fresh normal Tyrode’s solution and then imaged with the CoolLED pE-340^fura^ set up (CoolLED Ltd. Hempshire, UK). The acquisition rate was set to 10 Hz per ratio (R = F_340_/F_380_). Fura-2 ratios from myoplasmic areas of interest were calculated using Zeiss Zen software and then converted to resting free *Ca*^2+^ concentrations using an in situ calibration curve for Fura-2 AM according to the method of Grynkiewicz et al. [66], based on the formula:(3)$$\left[Ca^{2+}\right]=\frac{\beta \;K_{d}\;\left(R-R_{min}\right)}{\left(R_{max}-R\right)}$$
where *β K_d_*_(Fura-2)_ = 0.3, whereas *R_min_* and *R_max_* were determined for our set up as being equal to 0.3 and 7.4, respectively.

#### 4.5.6. Mitochondrial Calcium Uptake Measurement

Evaluation of the mitochondrial calcium uptake on single FDB fibers was conducted following Ainbinder and coworkers [67]. FDB fibers were loaded with 5 µM rhod-2 AM for 15 min at room temperature, which was washed out with normal Tyrode’s solution, containing 25 µM BTS to avoid contraction during the tetanic stimulation. We applied 5 consecutive tetanic stimulations (500 ms duration, 100 Hz) at supramaximal activating voltages with a pair of platinum electrodes placed close to the fiber of interest (S88 Stimulator, Grass Technologies, Warwick, RI, USA), while changes in mitochondrial calcium levels were recorded with confocal microscope (Zeiss 5 Live, Oberkochen, Germany). Time series x-y images (512 × 512 pixels, 0.5 ms/line) were taken at rest, following the 1st and 5th tetanus. Calculation of Rhod-2 fluorescence values originating from the mitochondria (*F_mito_*) were performed with the following method: a rectangle was drawn parallel to the longitudinal axis of the fiber and the fluorescence was calculated at the peaks (I-band fluorescence, representing mitochondria (*F_I-band_*)) and at troughs (A-band, fluorescence, baseline, *F_A-band_*). The normalized mitochondrial calcium uptake expressed as *F_mito_* was calculated with the equation:(4)$$F_{mito}=\frac{F_{I\text{-}band}-F_{A\text{-}band}}{F_{A\text{-}band}}$$

### 4.6. Statistical Analysis

Pooled data were expressed as mean±standard error of the mean (SEM). The differences between Cre^−/−^ and Cre^+/−^ mice were assessed using one-way analysis of variance (ANOVA) and all pair-wise Bonferroni’s multiple comparison method using the statistical program Prism (GraphPad Software, San Diego, CA, USA). *t*-test was used to test the significance and a *p* value of less than 0.05 was considered statistically significant.

## 5. Conclusions

Understanding the role of the skeletal endocannabinoid system in health and disease is of utmost importance to be able to address proper therapies in conditions where a defective ECS activity occurs. Here we give for the first time a functional and morphological characterization of a transgenic mouse model upon skeletal muscle specific Cnr1 genetic ablation. We did not address in depth the molecular mechanisms which may underlie some of the changes we describe here (e.g., altered muscle contractility and mitochondrial morphology); thus, we can only hypothesize about the possible players and the signaling pathways involved. We propose a possible link between the endocannabinoid system (more precisely CB1R), ATP production, and possibly ROS signaling; nevertheless, further investigations are needed to carefully evaluate these.

## Figures and Tables

**Figure 1 ijms-23-15653-f001:**
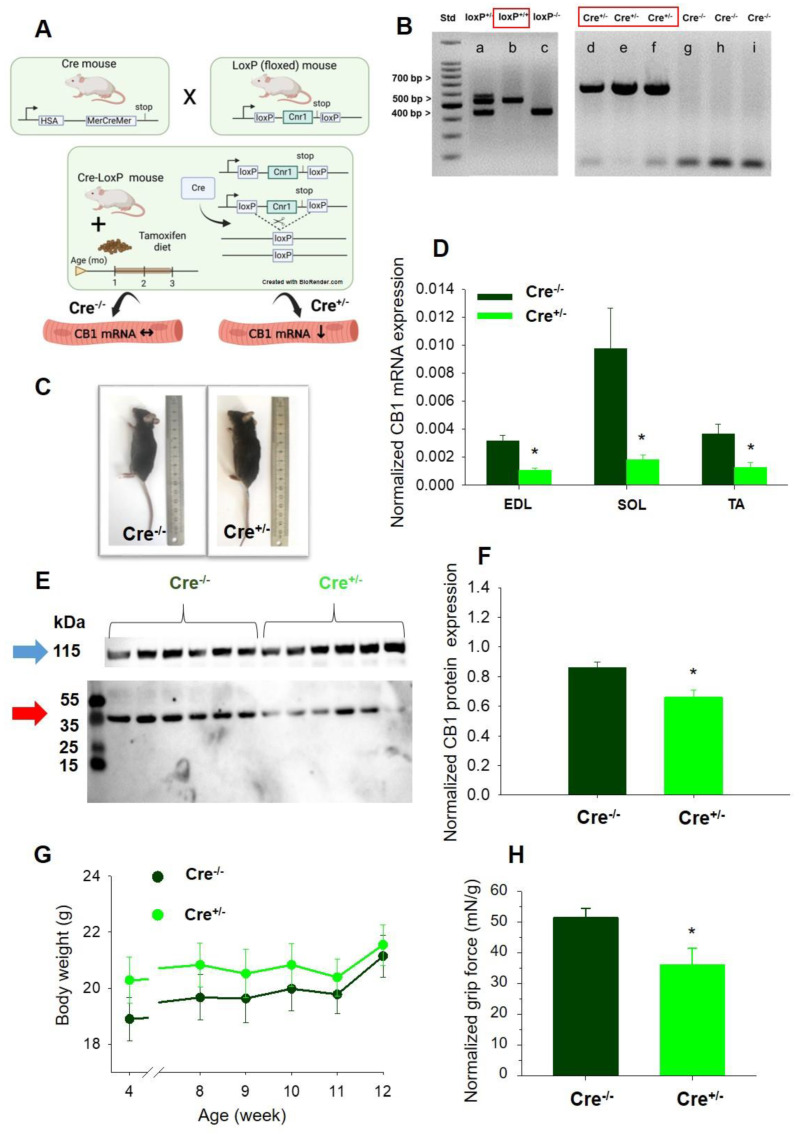
Skeletal muscle–specific knock–down of CB1R. (**A**) Schematic diagram of muscle-specific CB1R-knock-down breeding strategy. (**B**) Validation and identification of the loxP^+/−^ (a), loxP^+/+^ (b), loxP^−/−^ (c), Cre^+/−^ (d, e, f), and Cre^−/−^ (g, h, i) mice via agarose gel electrophoresis. (**C**) Images of Tamoxifen-fed Cre^−/−^ and Cre^+/−^ mice (all of them loxP^+/+^) at age of 3 months, following 2-month Tamoxifen diet period. The mice did not show any gross visible morphological defects and had normal behavior and movement. (**D**) Pooled data of the normalized CB1 mRNA expression levels in different muscle types (EDL, SOL, and TA) of Cre^−/−^ and Cre^+/−^ mice. 12 littermates (6 Cre^−/−^ and 6 Cre^+/−^ mice) were examined from 3 litters (breeding groups). PPIA (Peptidylprolyl Isomerase A) was used as internal control. (**E**) Representative Western Blot performed on TA muscles illustrating CB1 protein expression (red arrow, ~45 kDa) and α-actinin (blue arrow, ~115 kDa) that was used as normalizing gene. (**F**) Bar graph depicting average normalized CB1 protein level examined for both samples in 4 independent experiments. (**G**) Change in body weight during the 2-month Tamoxifen diet monitored for mixed gender Cre^−/−^ (*N* = 9) and Cre^+/−^ (*N* = 13 mice). (**H**) Grip force normalized to body weight in Cre^−/−^ and Cre^+/−^ mice (*N* = 4 for each group). * *p* < 0.05.

**Figure 2 ijms-23-15653-f002:**
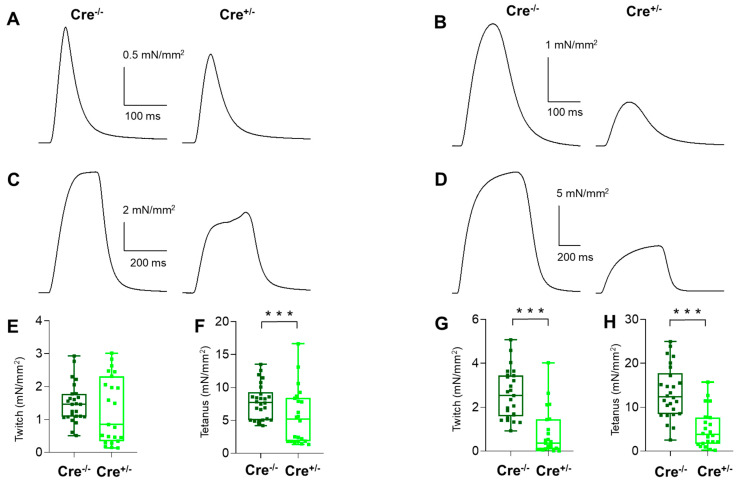
Altered isometric force in EDL and SOL muscle from Cre^+/−^ mice. Representative twitch (**A**,**B**) and tetanic force (**C**,**D**) transients in EDL (**A**,**C**) and in SOL (**B**,**D**). The force was normalized to the cross section of the muscle. Box plots show the individual data points for the peak twitch (**E**,**G**) and tetanic (**F**,**H**) force in EDL (**E**,**F**) and SOL (**G**,**H**) from Cre^−/−^ (*N* = 13 mice, *n* = 26/25 muscles) and Cre^+/−^ (*N* = 14 mice, *n* = 22 muscles). *** *p* < 0.001.

**Figure 3 ijms-23-15653-f003:**
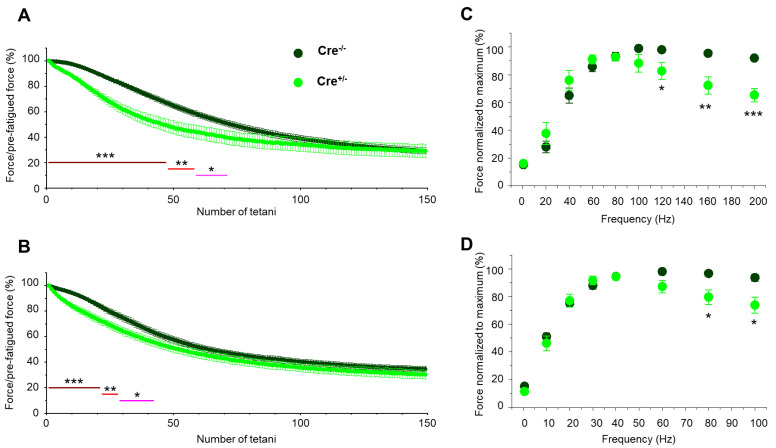
Altered fatigability and activation of EDL and SOL muscles in Cre^+/−^ mice. Fatigue in EDL (**A**) and in SOL (**B**) muscles from Cre^−/−^ (*N* = 13) and Cre^+/−^ (*N* = 14) mice. Magenta, red, and dark red line show the interval where Cre^+/−^ data is significantly different from Cre^−/−^ at *p* < 0.05, *p* < 0.01, and *p* < 0.001, respectively. Force-frequency relationship in EDL (**C**) and in SOL (**D**) muscles from Cre^−/−^ (*N* = 4) and Cre^+/−^ (*N* = 4) mice. * *p* < 0.05, ** *p* < 0.01, *** *p* < 0.001.

**Figure 4 ijms-23-15653-f004:**
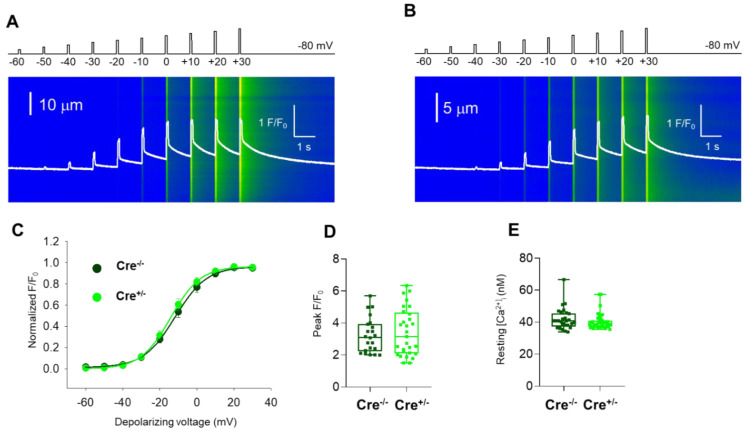
Unaffected resting intracellular calcium levels and release channel sensitivity to activation in fibers from Cre^+/−^ mice. Intracellular Ca^2+^ transients were recorded under whole-cell voltage clamp following Rhod-2 loading, with 100 ms-long gradually increasing membrane depolarization ranging between −60 mV and +30 mV, with 10 mV increments applied every 500 ms in single FDB fibers from Cre^−/−^ (**A**) and Cre^+/−^ (**B**) mice. Fluorescence was normalized to average resting F_0_(*x*). The white traces illustrate the spatially averaged F(*t*)/F_0_. (**C**) Data points represent the normalized maximal fluorescence at the given depolarization in 21 Cre^−/−^ and 27 Cre^+/−^ FDB muscle fibers obtained from 6 animals in each group. The voltage dependence of the normalized fluorescence was well fitted by a Boltzmann function using Equation (1). The mean values of parameters *V_50_* and *k* were not significantly different: *V_50_* (Cre^−/−^): −11.73 ± 2.1 mV, *k* (Cre^−/−^): 8.49 ± 0.6 mV, and *V_50_* (Cre^+/−^): −14.11 ± 2.4 mV, *k* (Cre^+/−^): 7.99 ± 0.6 mV. (**D**) Box plot distribution of the average peak of calcium transients elicited by single 100 ms-long depolarizations to +30 mV in Cre^−/−^ (*n* = 21 cells, *N* = 6 mice) and Cre^+/−^ fibers (*n* = 27 cells, *N* = 6 mice). The average peak F/F_0_ values were: 3.22 ± 0.23 and 3.41 ± 0.27. (**E**) Box plot representation of the resting intracellular calcium concentration as measured by Fura-2 ratiometric dye. The average resting calcium concentration values were: 41.8 ± 6.6 nM for Cre^−/−^ (*n* = 29 cells) and 40.3 ± 4.9 nM for Cre^+/−^ (*n* = 40 cells), respectively.

**Figure 5 ijms-23-15653-f005:**
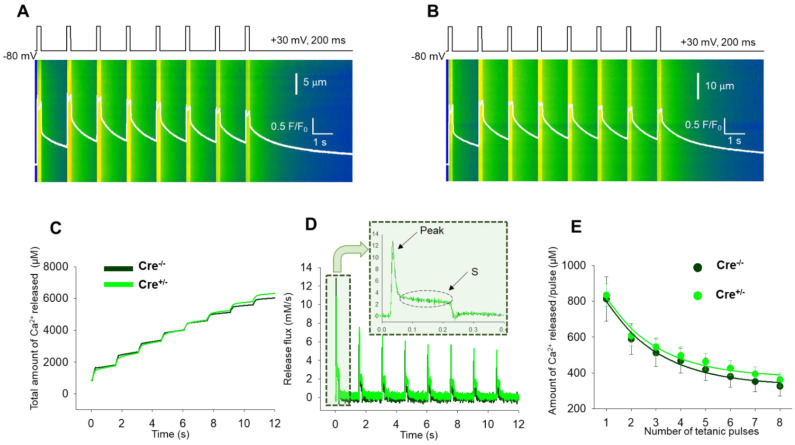
Unimpaired fatigability of Ca^2+^ release in FDB fibers from Cre^+/−^ mice. Representative line-scan images illustrating the fatigability of the FDB fibers in Cre^−/−^ (**A**) and Cre^+/−^ (**B**) fibers. The calcium transients were elicited by a train of depolarizing pulses to +30 mV with 1.5 s delay, lasting 200 ms each. The white trace is the temporal profile of the normalized fluorescence obtained by averaging 50 lines in the spatial domain normalized to average resting F_0_(x) values. (**C**) Evolution of the total amount of Ca^2+^ released during the train of depolarization for the fibers in (**A**,**B**). (**D**) Evolution of the release flux calculated from the cytosolic Ca^2+^ transients calculated for Cre^−/−^ and Cre^+/−^, respectively. Zoomed in view of the release flux illustrated for the first tetanic pulse of the series. The inset depicts two characteristic features of the release flux: the peak and the steady state (S). (**E**) The average amount of Ca^2+^ released per pulse for 19 Cre^−/−^ and 26 Cre^+/−^ fibers. A single exponential function was fitted to the points with the following parameters: y_0_ = 326.95, a = 744.03, b = 0.46 for Cre^−/−^ and y_0_ = 374.01, a = 714.90, b = 0.48 for Cre^+/−^.

**Figure 6 ijms-23-15653-f006:**
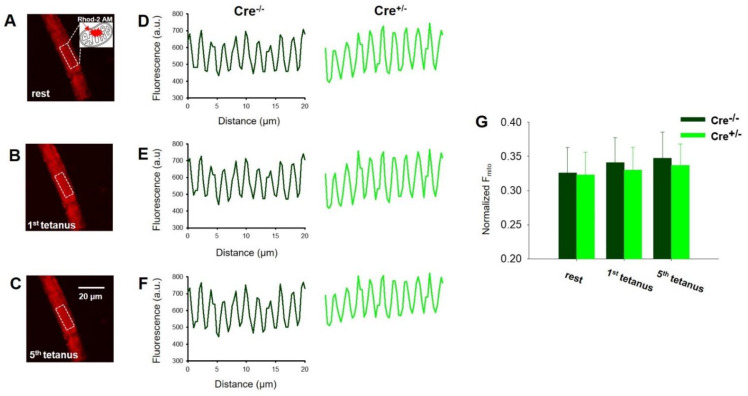
Unaltered activity dependent mitochondrial calcium uptake in FDB muscles from Cre^+/−^ mice. Representative confocal images of a Rhod-2 AM-loaded FDB fiber originated from a Cre^+/−^ mouse in resting condition (**A**), after the 1st (**B**), and after the 5th (**C**) tetanic stimulation. Calibration in panel (**C**) corresponds to both images shown in (**A**,**B**). (**D**–**F**) Rhod-2 fluorescence intensity profiles were calculated from similar dashed line delimitated areas as presented in panels (**A**–**C**) for both animal groups. (**G**) Average amplitude of Rhod-2 fluorescence intensity changes in resting condition, after the 1st and the 5th tetanic stimulation. The average (±SEM) change of normalized mitochondrial fluorescence (*F_mito_*) was calculated for *n* = 15–17 cells originated from 4–6 mice in each group based on Equation (4).

**Figure 7 ijms-23-15653-f007:**
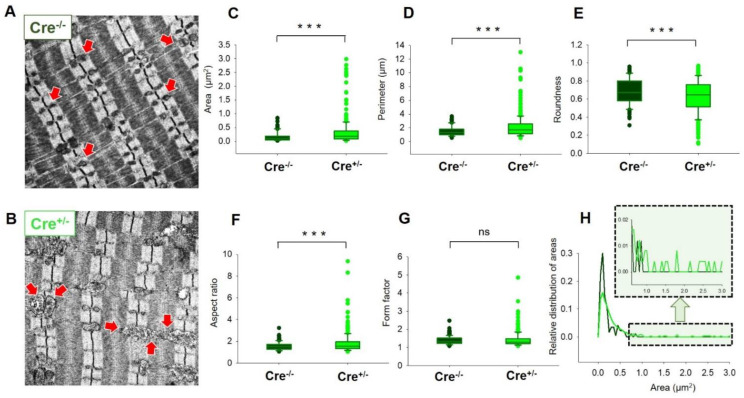
Significant mitochondrial morphology alterations in Cre^+/−^ muscles. Representative electron micrographs of myofibrils from longitudinal sections obtained on TA muscles samples from Cre^−/−^ (**A**) and Cre^+/−^ (**B**), depicting drastically altered mitochondrial morphology. The arrows indicate the mitochondria that are placed at the I band next to triads, always on the side closer to the Z line. Box plot representation of the morphological analysis of the individual mitochondria (*n* = 168 for Cre^−/−^ and *n* = 245 for Cre^+/−^ *N* = 3 mice/group) concerning the area (**C**), perimeter (**D**), roundness (**E**), aspect ratio (**F**), and form factor (**G**) determined from similar EM images as shown in panels (**A**,**B**). *** *p* < 0.001, ns: not significant. (**H**) Relative distribution of areas depicts a rightward shift in Cre^+/−^ muscles. The highlighted area illustrates a zoomed in view into this phenomenon.

**Figure 8 ijms-23-15653-f008:**
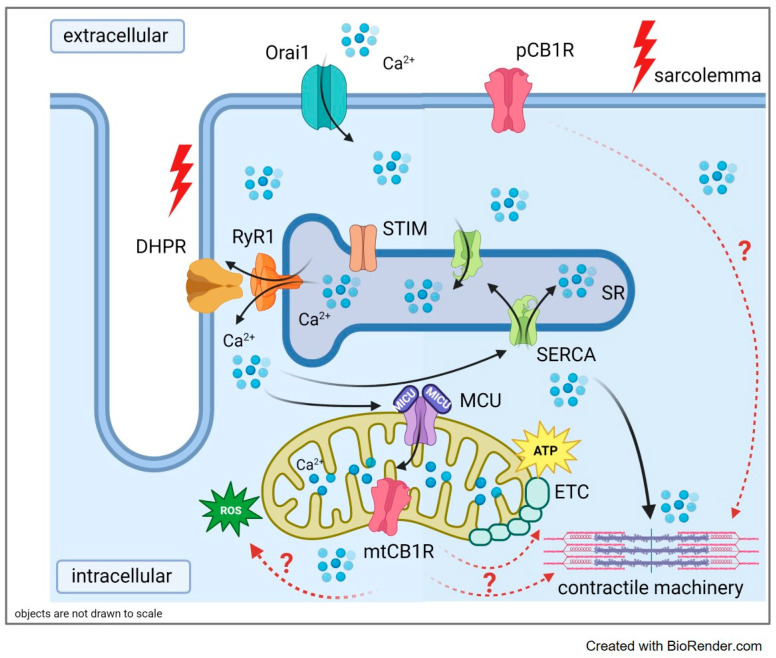
Summary of the proposed mechanism underlying hindered skeletal muscle function and ultrastructure by CB1 down-regulation. The arrows (black) indicate the path of calcium ions upon release from the sarcoplasmic reticulum (SR). Red arrows tackle the connection between the underlined players. Genetic ablation of CB1R leads to impaired contractility and altered mitochondrial morphology. Legend: pCB1R: plasma membrane cannabinoid receptor 1; mtCB1R: mitochondrial cannabinoid receptor 1; MCU (MICU1): mitochondrial calcium uniporter complex; ETC: electron transport chain; RyR1: ryanodine receptor type 1; DHPR: dihydropiridine receptor, STIM: stromal interaction molecule, Orai1: calcium release-activated channel 1; SERCA: sarco(endo)plasmic reticulum calcium pump; ROS: reactive oxygen species.

**Table 1 ijms-23-15653-t001:** Parameters of twitch and tetanus in EDL and SOL muscles.

	EDL				SOL			
	Twitch		Tetanus		Twitch		Tetanus	
	Cre^−/−^	Cre^+/−^	Cre^−/−^	Cre^+/−^	Cre^−/−^	Cre^+/−^	Cre^−/−^	Cre^+/−^
Number of muscles	26	22	26	22	25	22	25	22
Peak force (mN)	1.53 ± 0.09	0.80 ± 0.10 ***	8.08 ± 0.43	3.49 ± 0.42 ***	1.83 ± 0.12	0.57 ± 0.13 ***	9.41 ± 0.49	4.09 ± 0.69 ***
Force (mN/mm^2^)	1.49 ± 0.12	1.29 ± 0.21	7.84 ± 0.52	5.61 ± 0.91 *	2.60 ± 0.22	0.81 ± 0.22 ***	13.38 ± 1.16	5.09 ± 0.94 ***
TTP (ms)	37.1 ± 2.8	40.8 ± 1.8	209.9 ± 12.9	147.8 ± 13.4 **	109.1 ± 7.2	88.9 ± 4.2 *	517.3 ± 12.7	526.0 ± 2.6
HRT (ms)	32.5 ± 2.7	33.1 ± 1.3	69.2 ± 3.7	118.8 ± 11.1 ***	102.7 ± 10.9	110.6 ± 8.5	142.5 ± 5.9	98.9 ± 3.2 ***
Duration (ms)	246.1 ± 30.5	233.1 ± 14.4	376.4 ± 18.0	376.9 ± 14.2	401.2 ± 35.2	479.9 ± 46.4	834.0 ± 29.0	764.5 ± 13.6 *
Fatigue at 50th (%)			34.4 ± 2.3	52.0 ± 4.9 **			41.8 ± 2.6	48.6 ± 3.2
Fatigue at 100th (%)			60.9 ± 2.6	65.7 ± 5.4			59.5 ± 2.3	64.1 ± 3.3
Fatigue at 150th (%)			70.8 ± 2.5	71.0 ± 5.1			65.4 ± 2.2	69.7 ± 3.5
CSA (mm^2^)	1.07 ± 0.04	0.81 ± 0.06 **			0.80 ± 0.06	0.96 ± 0.07		
Muscle weight (mg)	12.8 ± 0.4	11.5 ± 0.3 *			11.0 ± 0.6	14.3 ± 0.5 ***		

*, ** and ***: significant difference from CRE- at *p* < 0.05, *p* < 0.01 and *p* < 0.001, respectively. TTP: time to peak; HRT: half relaxation time; CSA: cross sectional area.

**Table 2 ijms-23-15653-t002:** Sequences of Cre and CB1-loxP primers used for genotyping.

	Cre	CB1-loxP
Forward primer	5′-GCATGGTGGAGATCTTTGA-3′	5′-GCTGTCTCTGGTCCTCTTAAA-3′
Reverse primer	5′-CGACCGGCAAACGGACAGAAGC-3′	5′-GGTGTCACCTCTGAAAACAGA-3′

## Data Availability

Data will be made available on reasonable request.

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
