# Peer review of "Genetic Manipulation of CB1 Cannabinoid Receptors Reveals a Role in Maintaining Proper Skeletal Muscle Morphology and Function in Mice"

_ijms, 2022, doi:10.3390/ijms232415653_

Round 1
Reviewer 1 Report
Comments to the authors
The manuscript by Zoltan Singlar and others, entitled “Genetic manipulation of CB1R reveals a role in maintaining proper skeletal muscle morphology and function in mice” presents new insights about the role of CB1Rs in maintaining physiological muscle function and morphology. The authors clearly indicated that CB1 knock-down significantly alter the isometric force and the mitochondria morphology. The following comments need to be address by the authors:
1- In page 2, line 80: “To estimate the muscle-specific down-regulation of CB1 mRNA expression, total lysates of m. extensor digitorum longus (EDL), m. soleus (SOL), and m. tibialis anterior (TA) were examined and a significant partial down-regulation of CB1 mRNA levels (to 34%, 19%, and 34% respectively, compared to similar EDL, SOL and TA samples originated from Cre-/- mice.”
The authors may explain the significant down regulation in SOL (19%) compared to EDL and TA (34%)?
2- The author mentioned: “These findings imply that the smaller tetanic force observed in Cre+/- mice is not the consequence of a modified calcium handling and release channel activity.” Also, it is not an altered mitochondrial calcium handling that accounts for the smaller tetanic force observed in Cre+/- muscle fibers. Furthermore, the authors indicated a significant alteration of the mitochondrial morphology.”
Considering these findings, I believe that CB1 knock-down leads to significant alteration in the mitochondrial morphology which in turns lead to insufficient ATP production that finally results in reduced tetanic force and fatigability. This also could imply that CB1 KD leads to skeletal muscle aging!! Previous study indicated that CB1 receptors deletion leads to neuronal loss and brain aging!
This could be the possible mechanism, please consider further investigations! The authors may measure the mitochondrial functions and the ATP profile. Please check the following reports:
A. Sarcopenia is associated with complex changes in mitochondrial morphology that could interfere with mitochondrial function and mitophagy, and thus contribute to aging-related accumulation of mitochondrial dysfunction and sarcopenia. DOI: 10.18632/oncotarget.4235
B. Compromised mitochondrial morphology is correlated with insufficient ATP production. https://doi.org/10.1038/s41598-019-54159-1
C. Role of CB1 cannabinoid receptors on GABAergic neurons in brain aging. https://doi.org/10.1073/pnas.1016442108
3- There is a typo in the double quotation mark in line 200-201 at page 9, please correct.
4- Please give the full name of MICU protein in line 227 at page 10.
5- Reference 23 is missing the journal name and the DOI, please correct?
6- Reference 39, please correct the DOI.
Author Response
We thank the Reviewer for the careful reading of the manuscript and the constructive remarks. We have made the changes suggested to improve and clarify the manuscript. Please find below a detailed point-by-point response to all comments (reviewers’ comments in bold italics).
Answers to Reviewer #1
1- In page 2, line 80: “To estimate the muscle-specific down-regulation of CB1 mRNA expression, total lysates of m. extensor digitorum longus (EDL), m. soleus (SOL), and m. tibialis anterior (TA) were examined and a significant partial down-regulation of CB1 mRNA levels (to 34%, 19%, and 34% respectively, compared to similar EDL, SOL and TA samples originated from Cre-/- mice.”
The authors may explain the significant down regulation in SOL (19%) compared to EDL and TA (34%)?
We have revisited the data and even though the numbers suggest a significant down-regulation in the investigated muscle types in the Cre+/- mice compared to the Cre-/- specimens, no significant difference could be detected statistically within the Cre+/- samples between these muscle types. Now it is stated in the revised manuscript.
2- The author mentioned: “These findings imply that the smaller tetanic force observed in Cre+/- mice is not the consequence of a modified calcium handling and release channel activity.” Also, it is not an altered mitochondrial calcium handling that accounts for the smaller tetanic force observed in Cre+/- muscle fibers. Furthermore, the authors indicated a significant alteration of the mitochondrial morphology.”
Considering these findings, I believe that CB1 knock-down leads to significant alteration in the mitochondrial morphology which in turns lead to insufficient ATP production that finally results in reduced tetanic force and fatigability. This also could imply that CB1 KD leads to skeletal muscle aging!! Previous study indicated that CB1 receptors deletion leads to neuronal loss and brain aging!
This could be the possible mechanism, please consider further investigations! The authors may measure the mitochondrial functions and the ATP profile. Please check the following reports:
- Sarcopenia is associated with complex changes in mitochondrial morphology that could interfere with mitochondrial function and mitophagy, and thus contribute to aging-related accumulation of mitochondrial dysfunction and sarcopenia. DOI: 10.18632/oncotarget.4235
- Compromised mitochondrial morphology is correlated with insufficient ATP production. https://doi.org/10.1038/s41598-019-54159-1
- Role of CB1 cannabinoid receptors on GABAergic neurons in brain aging. https://doi.org/10.1073/pnas.1016442108
We thank the reviewer for pointing out these articles which are now cited in the discussion and have been also included in the reference list. As elaborated in the discussion and pointed out by the reviewer an insufficient, hindered ATP production due to altered mitochondrial morphology because of CB1 genetic manipulation could be a possible explanation behind the decreased force production we measured in vivo and in vitro.
We agree that further studies might shed light on this phenomenon. Our reasoning for not doing these experiments is the lack of available tools and equipment in our laboratory. One possibility to measure ATP profile in adult skeletal muscle fibers would be via the SeaHorse (Schuh et al. 2012 doi: 10.1152/ajpregu.00229.2011) or Oroboros (O2k-FluoRespirometer) systems. Another possibility would be the usage of ATP sensitive microelectrodes as in Buvinic et al. 2009 (doi: 10.1074/jbc.M109.057315) but again, unfortunately these tools are not available to us. Lastly, luminescent ATP kits are also a possible approach, however these are mainly designed to give adequate results on cell suspension samples and not on skeletal muscle cells.
From the patch clamp experiments performed on FDB cells we have calculated the maximum rate of Ca2+ removal (PVmax values) and even though there was a small decrease in Cre+/- samples compared to Cre-/- this was not significant (3617±297 vs 3429±263 µM/sec). As there is no reason to believe that SERCA1 expression would be altered in our mouse model the slight but not significant decrease in pumping rate could be associated with a reduction in ATP production in Cre+/- mice.
3- There is a typo in the double quotation mark in line 200-201 at page 9, please correct.
Thank you, this was now corrected.
4- Please give the full name of MICU protein in line 227 at page 10.
Done.
5- Reference 23 is missing the journal name and the DOI, please correct?
Done.
6- Reference 9, please correct the DOI.
Done.

Reviewer 2 Report
Manuscript ID: ijms-2041570
Genetic manipulation of CB1R reveals a role in maintaining proper skeletal muscle morphology and function in mice.
The manuscript by Singlár and coworkers reports on studies aimed at characterizing the functional and morphological role of the cannabinoid receptor1 (CB1R) in a knock-down mouse model. Authors found decreased grip force production and more propensity to fatigue in Cre+/- muscles. These alterations could be produced by a disruption in the calcium homeostasis of these muscle cells, so the authors investigated the impact of this ablation. Still, none of these phenotypic characteristics were explained by alterations in the excitation-contraction coupling process; but interestingly, ultrastructural changes in mitochondrial morphology were found.
I have a few concerns that should be addressed before proceeding with consideration for publication in IJMS.
- All electrophysiological experiments were carried out on FDB fibers, whereas the biochemical studies were carried out on three different muscles (EDL, Soleus, and TA). Therefore, representative blots should be performed to show that the same changes in the expression of CB1 protein also occur in FDB muscles.
- When describing the susceptibility to fatigue in Fig.3, authors should not compare "between muscles" (line141) if the fatigue protocols applied were not the same (see methods). It is correct, though, to compare phenotypes in the same muscles (e.g., Cre+/- and Cre-/- in soleus) as also done in this section.
- Observing the results in Fig. 4, the parameters reported are incomplete. It would be important to report in a "table" what happens with Qmax (and the rest of the parameters) in this set of experiments.
- Additionally, when comparing panels "A" and "B," we observe that calcium release starts occurring before in Cre+/- (compare amplitudes at -10 mV, for instance). Values reported in panel "C" present a minimal variation according to what is shown in these two panels. Authors need to be careful when analyzing this data; it is not the same to average each point at a particular voltage and then adjust the result to a Boltzmann than to average a set of Boltzmanns; this might be overkilling the differences if there are some.
- Authors should not conclude that "CB1 genetic manipulation does not alter the ECC machinery…" but leave it as a still open question and conclude that this is happening under these conditions and stimulation protocols.
- Fig.5C shows a smaller "Total amount of calcium released" in Cre+/- after several "tetanic" stimulations. Additionally, in Fig.5D, the peak of calcium flux is smaller in these muscle fibers. Couldn't this lower release be used as a partial explanation for the reduction in force production? Could not this also be related to a possible change in V50 and/or Qmax (as stated in the previous point)?
- It is difficult to observe differences in mitochondrial calcium only after 5 tetanic stimulations (Fig.6G). Why didn't the authors perform these measurements when the fiber was considered fatigued? It is strange that finding such a difference in the size of mitochondria (Fig.7), there is no change in the regulation of calcium.
- Authors conclude that ROS could affect the contractile machinery and therefore alter force production. However, ROS also has been observed to affect RyR activity and the activity of other ECC-related proteins. In addition, it has been previously shown that the calcium release event alteration does exist, as stated in ref. 28 (Olaf et al., 2016). It is important to clarify the mechanisms that lead to decreased force production that the authors propose.
Minor changes:
- Authors should consider modifying the conclusions in the abstract section, as they did not find any alteration in "calcium fluxes across cellular membranes," and adjust this section to their conclusions.
- Similarly, as pointed out in the previous comment, the introduction section should be improved to clarify the focus of this important work; the conclusions are too ambiguous.
- References to previous works should be revised; e.g., in the introduction, authors used a 2022 reference (ref. 23, line 52) to a phenomenon described decades before.
- The size of panel "A" in Fig.1 is too small and with low resolution; please fix this.
Author Response
Answers to the Reviewer’s comments.
We thank the Reviewer for the careful reading of the manuscript and the constructive remarks. We have made the changes suggested to improve and clarify the manuscript. Please find below a detailed point-by-point response to all comments (reviewers’ comments in bold italics).
Answers to Reviewer #2
All electrophysiological experiments were carried out on FDB fibers, whereas the biochemical studies were carried out on three different muscles (EDL, Soleus, and TA). Therefore, representative blots should be performed to show that the same changes in the expression of CB1 protein also occur in FDB muscles.
We thank the reviewer for this comment. We would have proceeded this way if muscles would have been available for molecular biology studies, such as Western Blot. However, since we had a limited number of tamoxifen fed specimens and the in vitro force measurements were performed on EDL (fast glycolytic type muscle) and SOL (slow oxidative type muscle), whereas other functional measurements (patch clamp and mitochondrial calcium uptake) were carried out on FDBs (considered as a fast twitch muscle), we had to make the decision to employ for Western Blotting a mixed composition muscle type that is the TA, which is big enough to prepare sufficient protein samples. The TA muscle is overall considered as a fast contracting muscle, which, according to published studies (e.g. Augusto et al. 2004) is composed of about 5% type-IIA, 35% type-IIX, and 60% type-IIB fibers.
When describing the susceptibility to fatigue in Fig.3, authors should not compare "between muscles" (line141) if the fatigue protocols applied were not the same (see methods). It is correct, though, to compare phenotypes in the same muscles (e.g., Cre+/- and Cre-/- in soleus) as also done in this section.
We would like to apologize if the comparison of speed of fatigue between genotypes was not completely clear. We modified the text to solve this discrepancy.
Observing the results in Fig. 4, the parameters reported are incomplete. It would be important to report in a "table" what happens with Qmax (and the rest of the parameters) in this set of experiments.
Thank you for this comment. Camax values are now given in the legend of Figure 4 and we have included a supplementary figure (Figure S1) to show the normalized Camax values for fibers from each sample.
Additionally, when comparing panels "A" and "B," we observe that calcium release starts occurring before in Cre+/- (compare amplitudes at -10 mV, for instance). Values reported in panel "C" present a minimal variation according to what is shown in these two panels. Authors need to be careful when analyzing this data; it is not the same to average each point at a particular voltage and then adjust the result to a Boltzmann than to average a set of Boltzmanns; this might be overkilling the differences if there are some.
Thank you for pointing this out. We have now selected a new line scan image to better illustrate the similarities between the two strains in terms of depolarization induced calcium transients (Figure 4A and B). Also, we redraw the box plots in panels D and E of the same figure.
Authors should not conclude that "CB1 genetic manipulation does not alter the ECC machinery…" but leave it as a still open question and conclude that this is happening under these conditions and stimulation protocols.
We have modified the section’s conclusion taking into account this comment.
Fig.5C shows a smaller "Total amount of calcium released" in Cre+/- after several "tetanic" stimulations. Additionally, in Fig.5D, the peak of calcium flux is smaller in these muscle fibers. Couldn't this lower release be used as a partial explanation for the reduction in force production? Could not this also be related to a possible change in V50 and/or Qmax (as stated in the previous point)?
We have chosen a different representative measurement for Figure 5C which aligns better with the averages as further detailed in Table S1. On the other hand, the peak of calcium fluxes turns out to be slightly smaller, but this is not significant between the fibers studied for the two samples.
It is difficult to observe differences in mitochondrial calcium only after 5 tetanic stimulations (Fig.6G). Why didrage n't the authors perform these measurements when the fiber was considered fatigued? It is strange that finding such a difference in the size of mitochondria (Fig.7), there is no change in the regulation of calcium.
The protocol we applied here to follow the mitochondrial calcium uptake following tetanic stimulation has been already published by us (Sztretye et al. 2020, doi: 10.3390/antiox9020098. and Singlár et al. 2021 doi: 10.3390/antiox10091415) and others (Ainbinder et al. 2015, doi: 10.3390/antiox10091415). Thus we decided not to change this protocol and employ it here as done previously.
On the other hand, when one looks at the average parameters of the voltage induced calcium transients and the calculated release fluxes during tetanic stimulation (Table S1) no significant differences could be seen between the two strains. We have also recorded a different protocol as described in our previous report (see Fig 6 in Sztretye et al. 2017, DOI: 10.1016/j.bpj.2017.09.023). Briefly, 50 consecutive small depolarizing pulses (20 ms) from -80 mV to +30 mV were applied on FDB fibers and when comparing the ratios of the peak fluorescence of the transients (the 25th and the 50th transients were normalized to the first transient of the series) we could see a small but insignificant decrease of the fluorescence ratio in Cre+/- fibers (0.79±0.03 and 0.66±0.03, n=28) compared to the similar values in Cre-/- (0.85±0.02 and 0.72±0.03, n=19 fibers). This data correlated with the decreased force production (Figure 2 left panel) and fatigue (Figure 3A) measured in vitro on EDL muscles.
Authors conclude that ROS could affect the contractile machinery and therefore alter force production. However, ROS also has been observed to affect RyR activity and the activity of other ECC-related proteins. In addition, it has been previously shown that the calcium release event alteration does exist, as stated in ref. 28 (Olaf et al., 2016). It is important to clarify the mechanisms that lead to decreased force production that the authors propose.
We understand that concerns of the reviewer regarding the possible involvement of ROS in the altered contractility we describe in this mouse model. Our goal was to characterize this newly generated mouse model and certain findings (i.e. altered force production and mitochondrial morphology) were surprising to us as well. We can only hypothesize about the possible mechanism(s) behind these findings. One possibility we think might be at work is a faulty ROS generation, however we do not have any proof for that as we did not perform ROS measurements here. Further studies are needed to elucidate this.
Minor changes:
Authors should consider modifying the conclusions in the abstract section, as they did not find any alteration in "calcium fluxes across cellular membranes," and adjust this section to their conclusions.
In the last sentence of the abstract we simply wanted to reinforce the importance of endocannabinoid system in skeletal muscle and more importantly in muscular disorders as it has been shown that in certain muscular dystrophies (i.e. Duchenne muscular dystrophy) increased endocannabinoid levels and aberrant calcium fluxes across cellular membranes were described (see Iannotti et al. 2018). These findings justify studying the importance and understanding ECS in skeletal muscle both in health and disease. Nevertheless, in response to the reviewers request we have removed the statement regarding the calcium fluxes across cellular membranes.
Similarly, as pointed out in the previous comment, the introduction section should be improved to clarify the focus of this important work; the conclusions are too ambiguous.
References to previous works should be revised; e.g., in the introduction, authors used a 2022 reference (ref. 23, line 52) to a phenomenon described decades before.
Thanks for pointing this out, new references have been added.
The size of panel "A" in Fig.1 is too small and with low resolution; please fix this.
Panel A of Figure 1 have been saved with high resolution and inserted. Thanks for drawing our attention to this.

Reviewer 3 Report
Singlár et al., investigated the role of CB1 in regulating skeletal muscle function. The topic is interesting. CB1 knockdown did not significantly affect muscle function, although changes of mitochondrial morphology were observed.
Please make it clear that the gene was knockdown is CB1 not CB1R. It is very confusing in the abstract, because it seems that you treat CB1 and CB1R as the same. For example, L20 and L22. Please check the whole manuscript and make sure it is consistent and clear.
Why did you generate the CB1 heterogenous mice rather than CB1 homogenous mice?
The variation is so big in Figure1 G. It might be due the mixed sex. Can you separate the male mice from the female and show the body weight with male and female individually? Besides, did you check the food intake?
Western data showed that the variation of CB1 knockdown is big (Figure 1 E). Is it possible this contributes to the variation of body weight?
Did you check skeletal muscle mass or the histology of skeletal muscle?
Figure 3 change "an" to "and"
As you showed in your proposed models, STIM1, Orai1 and other proteins are responsible for Ca2+ homeostasis. Did you check the protein abundance of these proteins? It is possible some of them increase to compensate for the loss of CB1.
Although the mechanism is not clear in the current study, it is helpful to further test if the mitochondria function is altered, since you observed the changes of mitochondria morphology.
Author Response
Answers to the Reviewer’s comments.
We thank the Reviewer for the careful reading of the manuscript and the constructive remarks. We have made the changes suggested to improve and clarify the manuscript. Please find below a detailed point-by-point response to all comments (reviewers’ comments in bold italics).
Answers to Reviewer #3
Please make it clear that the gene was knockdown is CB1 not CB1R. It is very confusing in the abstract, because it seems that you treat CB1 and CB1R as the same. For example, L20 and L22. Please check the whole manuscript and make sure it is consistent and clear.
We apologize for this confusing approach; the manuscript has been reviewed and corrections were done where needed according to the reviewer’s suggestion.
Why did you generate the CB1 heterogenous mice rather than CB1 homogenous mice?
Our goal was not to generate CB1 heterogenous mice. During the multiple breeding steps as detailed in Methods CB1flox/wt mice also resulted because of Mendelion genetics but these specimens were not used for subsequent experiments. Genotyping was always performed to distinguish between CB1flox/wt and CB1flox/flox. The latter were used for the breeding with the Cre positive mice as presented in Figure 1A. The Cre genotype was kept in heterozygous form so that breeding resulted in Cre positive CB1 floxed mice together with littermate controls.
The variation is so big in Figure1 G. It might be due the mixed sex. Can you separate the male mice from the female and show the body weight with male and female individually? Besides, did you check the food intake?
To answer the reviewer’s inquiry, we did not monitor the food intake per se for each individual mouse. For the experiments described in the present paper the mice were housed in groups of 4-5 mice/cage and had ad libitum access to food and water. However, we looked into the sex differences and body weight change in males and females and we generated two graphs to illustrate this (please the word file).
To begin with, at 4 weeks of age (the time when the mice were started on Tamoxifen diet) there was a slight variation in the average starting weight of the animals (20.21 g vs 22.1 g for males and 16.26 g vs 18.16 g for females) in Cre-/- and Cre+/-, respectively. During the 2 months of Tamoxifen diet there was a slight weight increase observed which turned out to be significantly less in the Cre+/- males at 12 weeks of age (*p<0.02).
Western data showed that the variation of CB1 knockdown is big (Figure 1 E). Is it possible this contributes to the variation of body weight?
We have checked the correlation between the CB1 down-regulation and body weight change but there was no correlation found between them (r2=0.28, p>0.5).
Did you check skeletal muscle mass or the histology of skeletal muscle?
The average weight of EDL and SOL muscles was measured prior to the in vitro force measurement experiments and the values are given in Table 1. We observed significantly lower muscle weight in EDL (fast type muscle), while significantly higher muscle weight in SOL (slow type muscle) from Cre+/- mice comparing to Cre-/- mice.
Furthermore, we have now analyzed the cross-sectional area of myofibrils in EM micrographs from TA muscles and found significantly lower area (3.46±0.15 µm2 vs 1.99±0.02 µm2, p<0.001) and perimeter (5.52±0.13 µm2 vs 4.05±0.03 µm2, p<0.001) of myofibrils in Cre+/- mice (N=3, n=208) comparing to Cre-/- mice (N=3, n=459).
Also, per the inquiry of the reviewer within the last few days we performed muscle histology and analysis on TA muscles originated from one Cre+/- and one Cre-/- animal. We employed hematoxylin-eosin staining on cryomatrix embedded slices of TA muscles. We quantified the area and perimeter of the muscle fibers and significant alterations were found following Cnr1 gene ablation. The area of the fibers was significantly decreased from 1503.08±41.25 in Cre-/- (n=215) to 1115±19.10 µm2 in Cre+/- (n=315) with p<0.001. In a similar fashion the perimeter of the fibers was drastically decreased from 157.55±2.35 µm2 to 136.41±1.43 µm2 with p<0.001).
Figure 3 change "an" to "and"
Done.
As you showed in your proposed models, STIM1, Orai1 and other proteins are responsible for Ca2+ homeostasis. Did you check the protein abundance of these proteins? It is possible some of them increase to compensate for the loss of CB1.
Recently, we have initiated a new set of experiments in collaboration with colleagues in Vienna. To begin we have performed Western Blot analysis on TA muscles to check the protein levels of the two main proteins involved in SOCE. Based on two independent experiments we did not find significant alterations in the levels of STIM1 and Orai1 in TA muscles originated from Cre-/- and Cre+/- samples, but have seen a tendency. Further experiments are needed to elucidate a possible change in protein expression as well as SOCE function. We believe that these data should be presented in a new manuscript.
Although the mechanism is not clear in the current study, it is helpful to further test if the mitochondria function is altered, since you observed the changes of mitochondria morphology.
Reviewer #1 had similar concerns and we can only reinstate the answer we gave them.
We agree that further studies might shed light on this phenomenon. Our reasoning for not doing these experiments is the lack of available tools and equipment in our laboratory. One possibility to measure ATP profile in adult skeletal muscle fibers would be via the SeaHorse (Schuh et al. 2012 doi: 10.1152/ajpregu.00229.2011) or Oroboros (O2k-FluoRespirometer) systems. Another possibility would be the usage of ATP sensitive microelectrodes as in Buvinic et al. 2009 (doi: 10.1074/jbc.M109.057315) but again, unfortunately these tools are not available to us. Lastly, luminescent ATP kits are also a possible approach, however these are mainly designed to give adequate results on cell suspension samples and not on skeletal muscle cells.

Round 2
Reviewer 1 Report
I have no further comments.
Reviewer 2 Report
· I advise the authors to end the introduction with a summary of their findings.
· In line 82, authors should specify that there are no differences in down-regulation of CB1 mRNA levels between muscles only in “cre-/+.”
· It is wrong to define the y-axis as “fatigue”; it should be defined as a decrease in “force” (i.e., % force/pre-fatigued force). Fatigue is defined as the decrease of force to a certain % in a specific period of time; therefore, what changes in this plot -Fig.3-is the rate of decrease in force generation.
· In line 142, fatigue is not “higher”; I would describe this as a faster decrease in force production.
· Fig 4 was not included in this version. However, authors shouldn’t refer to the max value as "Camax" unless you are calculating calcium values; in this case, it should be (F/F0)max.
· Basal [Ca2+] seems to be low; is this something reported before in this mouse model? Are these values found in regular WT muscle fibers in your lab? Maybe authors should check their Fura-2 calibration.
· I’m glad you fixed the boxplots; now we can see the data distribution.
Author Response
Answers to the Reviewer’s comments.
We thank the Reviewer for the careful reading of the manuscript and the constructive remarks. We have made the changes suggested to improve and clarify the manuscript. Please find below a detailed point-by-point response to all comments (reviewers’ comments in bold italics).
Answers to Reviewer #2
I advise the authors to end the introduction with a summary of their findings.
Done.
- In line 82, authors should specify that there are no differences in down-regulation of CB1 mRNA levels between muscles only in “cre-/+.”
Done.
- It is wrong to define the y-axis as “fatigue”; it should be defined as a decrease in “force” (i.e., % force/pre-fatigued force). Fatigue is defined as the decrease of force to a certain % in a specific period of time; therefore, what changes in this plot -Fig.3-is the rate of decrease in force generation.
We have corrected the-axis on Figure 3A and B.
- In line 142, fatigue is not “higher”; I would describe this as a faster decrease in force production.
Corrected.
- Fig 4 was not included in this version. However, authors shouldn’t refer to the max value as "Camax" unless you are calculating calcium values; in this case, it should be (F/F0)max.
Based on the Reviewer’s previous comment we have revisited the data and we have converted the F/F0 values to calcium to determine Camax taking into account the parameters detailed in the Methods sections of the manuscript (i.e. Kd (Rhod-2) = 1.58 μM, kON=0.07 µM-1 ms-1, and kOFF = 130 s-1). We have included a new supplementary figure (Figure S1) to show the Boltzmann distribution of the Camax values.
Figure 4 panels C and D stand to show the distribution of the F/F0 values and not the calcium. We had removed the reference to Camax from the figure legend and transferred it into the Figure S1 legend.
Basal [Ca2+] seems to be low; is this something reported before in this mouse model? Are these values found in regular WT muscle fibers in your lab? Maybe authors should check their Fura-2 calibration.
There are many reports in the literature tackling the systemic CB1 knock-out mice (https://doi.org/10.1073/pnas.1406728111; https://doi.org/10.3389/fphys.2016.00476). The mouse model we characterize here (Tamoxifen inducible skeletal muscle specific CB1 knock –down) is unique, exclusively generated by us and has not been studied before therefore we are the first to measure and report calcium concentrations (resting or upon voltage activation). Our CoolLed setup which uses a diode light source to excite Fura-2 to measure resting calcium concentrations has been calibrated and we are confident that the values are correct. (Rmin=0.3; Rmax=7.4; Kd*β=0.3 µM). We do acknowledge however, that this calibration was done in solution and not in the fiber and there are reports that the Kd of Fura-2 is higher intracellularly than in the solution. Thus, our values could slightly be underestimating the actual intracellular calcium concentration values.
- I’m glad you fixed the boxplots; now we can see the data distribution.
Sure. It is indeed a better, more suitable illustration of data distribution.

Reviewer 3 Report
The manuscript has been improved. It is very interesting to see the different response in oxidative fiber and glycolytic fiber. CB1R knockdown results in increased muscle mass in soleus and decreased muscle mass in EDL. Besides, increased soleus muscle mass is accompanied with reduced muscle force. Please discuss more about the muscle fiber type specific response in the discussion. As you know, enhanced exercise affects muscle fiber type shifting, it is good to mention how exercise influences the gene and protein of CB1R in the discussion or in the introduction. This may help explain the muscle fiber type specific response to CB1R knockdown. I assume you may see difference in body weight between WT and CB1R knockout mice, if the knockout mice are viable.
Author Response
Answers to the Reviewer’s comments.
We thank the Reviewer for the careful reading of the manuscript and the constructive remarks. We have made the changes suggested to improve and clarify the manuscript. Please find below a detailed point-by-point response to all comments (reviewers’ comments in bold italics).
Answers to Reviewer #3
The manuscript has been improved. It is very interesting to see the different response in oxidative fiber and glycolytic fiber. CB1R knockdown results in increased muscle mass in soleus and decreased muscle mass in EDL. Besides, increased soleus muscle mass is accompanied with reduced muscle force.
Please discuss more about the muscle fiber type specific response in the discussion. As you know, enhanced exercise affects muscle fiber type shifting, it is good to mention how exercise influences the gene and protein of CB1R in the discussion or in the introduction. This may help explain the muscle fiber type specific response to CB1R knockdown. I assume you may see difference in body weight between WT and CB1R knockout mice, if the knockout mice are viable.
Thank you for this comment. We have extensively reworked the Discussion section of the manuscript to further detail the connection between exercise and muscle fiber type switch.
